# The Sea Slug *Doriopsilla areolata* Bergh, 1880 (Mollusca, Gastropoda) in the Mediterranean Sea: Another Case of Cryptic Diversity

**Giulia Furfaro** [1,*], **Christopher Schreier** [1], **Egidio Trainito** [2], **Miquel Pontes** [3], **Enric Madrenas** [3], **Pascal Girard** [4] **and Paolo Mariottini** [5]

1 Department of Biological and Environmental Sciences and Technologie—DiSTeBA, University of Salento, Via Prov. le Lecce-Monteroni, 73100 Lecce, Italy; christopher.schreier@studenti.unisalento.it
2 Marine Protected Area 'Tavolara-Punta Coda Cavallo', 07026 Olbia, Italy; et@egidiotrainito.it
3 VIMAR (Vida Marina), C/Rocafort 246, 08029 Barcelona, Spain; miquelpontes@gmail.com (M.P.); enricmadrenas@gmail.com (E.M.)
4 Loeilduplongeur, Mauguio, 34130 Montpellier, France; a.pascalgirard@gmail.com
5 Department of Science, University of "Roma Tre", Viale G. Marconi 446, 00146 Rome, Italy; paolo.mariottini@uniroma3.it
* Correspondence: giulia.furfaro@unisalento.it; Tel.: +39-0832-29-8660

**Abstract:** The Mediterranean Sea diversity is still far from being fully disclosed. Marine Heterobranchia are one of the most paradigmatic species-rich groups, with many recent systematic studies revealing the high density of new, cryptic, and endemic species occurring in the Mediterranean basin. In this study, sea slug *Doriopsilla areolata*, which was considered until today one of the most widespread nudibranchs worldwide, was investigated using a molecular approach to compare Mediterranean and Atlantic populations for the first time. The molecular analyses involved three different molecular markers, the two mitochondrial COI and 16S, and the nuclear H3 gene. The results revealed a complex of species within *D. areolata* that indeed consists of three potentially species, two of which are endemic to the Mediterranean Sea: *Doriopsilla areolata*, which is distributed in the Adriatic Sea (the type locality of the former species), *D. rarispinosa*, which occurs in the Western Mediterranean basin and along the Tunisian coast, and one additional Atlantic species here provisionally defined as *Doriopsilla* sp. 1. This study helps to unveil another case of cryptic diversity within Mediterranean Heterobranchia and to increase the knowledge on *Doriopsilla* genus diversity.

**Keywords:** Heterobranchia; Nudibranchia; phylogeny; evolution; species identification; species complex

## 1. Introduction

Molluscs form the second largest phylum after arthropods with approximately 100,000 described species and potentially still 100,000 species to be described [1] with nearly 75% of species belonging to the Class Gastropoda [2]. Nudibranchs are the largest marine suborder within the Heterobranchia clade, and they are characterized by having lost their shell secondarily [3]. *Doriopsilla areolata* Bergh, 1880, is a radula-less nudibranch belonging to the family Dendrodorididae O'Donoghue, 1924 [4–6]. The absence of the radula is a characteristic feature within the family Dendrodorididae, and the reason they were classified in the group Porostomata Bergh, 1878 [5] together with the Phyllidiidae Rafinesque, 1814. However, the validity of Porostomata group has been discussed for several years and still needs to be better clarified [5,7–10]. Historically, the species within Dendrodorididae and Phyllidiidae are distinguished on the base of morphological diagnostic features [5,10] as the first ones are similar in appearance to other doridoidea, with their gills on the dorsum [5,10], while Phyllidiidae have their secondary gills typically located in a ventral/lateral position [5,10]. Two genera of the family Dendrodorididae are present in the Mediterranean:

*Doriopsilla* Bergh 1880 and *Dendrodoris* Ehrenberg 1831 [4,10]. Their morphological and molecular separations are currently clearly defined and supported [5,10–13].

The genus *Doriopsilla* groups 24 accepted taxa, with *D. areolata* as the type species [14]. The distribution range of the different species included in the mentioned genus spans almost worldwide [10,15], as they occur in the shallow coastal waters of the Pacific coast of North America, Australia, Indo-Pacific, Persian Gulf, South and West Africa, Mediterranean Sea, European Eastern Atlantic, North American Western Atlantic, and Caribbean Sea [10,15–17]. To date, three species are reported in the Mediterranean Sea: *Doriopsilla areolata* Bergh, 1880 [10,15]; *D. evanae* Ballesteros and Ortea, 1980 [18]; and *D. pelseneeri* d'Oliveira, 1895 [11] as *Dendrodoris minima* [15,19–24]. Some authors [15] hypothesized *D. evanae* as a possible synonym of *D. areolata*; however, they are currently regarded as distinct valid species. Other synonymized names are known for *D. areolata*, including *D. fedalae* Pruvot-Fol, 1953; *D. pusilla* Pruvot-Fol, 1951; *D. rarispinosa* Pruvot-Fol, 1951; and *Doris reticulata* Schulz in Philippi, 1836: The latter is considered as invalid, being the junior homonym of *Doris reticulata* Quoy & Gaimard, 1832 [14]. A possibility that *D. areolata* hides a complex of cryptic species has also been suggested [25,26].

*Doriopsilla areolata* was described for the first time based on two specimens from Hvar Island in the Adriatic Sea (Eastern Mediterranean Sea) [27], but its geographic distribution range is currently deemed from the Eastern Atlantic Ocean (Spain to Angola), the Mediterranean Sea, and the Caribbean Sea [6,15,28]. *Doriopsilla areolata* can feed on corneous sponges [4,5] by secreting digestive enzymes directly onto the sponge and sucking up predigested organic matter by means of a pore-like transformed mouth and a foregut modified into a suctorial tube [4,5,29,30]. The body color pattern shows a high intraspecific variability varying from yellow in smaller *D. areolata* individuals to light brown or grayish in larger individuals [15]. Interestingly, different morphotypes can be observed within the species, with individuals showing patterns with small white spots or rings and lines on the dorsum and many tubercles that can be flat or rounded [15]. *Doriopsilla areolata* was split into three distinct subspecies based on their different geographical distributions [15]: *D. areolata areolata* Bergh, 1880, in the Mediterranean Sea and the Eastern Atlantic Ocean (from the coasts of Spain to the Cape Verde Islands), *D. areolata albolineata* Edmunds, 1968, along the southwestern coasts of Africa (from Ghana to Angola and the island of São Tomé) [15,31], and *D. areolata nigrolineata* Meyer, 1977 along the Caribbean coast of Panama [15,32]. The institution of subspecies was encouraged by the widespread distribution of *D. areolata*, supported by its planktotrophic developmental strategy that could have led to a higher dispersal potential compared to species with lecithotrophic or direct development [6,33]. However, molecular investigations based on the mitochondrial 16S and the nuclear H3 molecular markers [28,33] put in doubt the validity of the division of *D. areolata* into different subspecies. To confirm these speculations, the need to repeat the analyses with a larger sample size and additional and more informative genes, such as the barcoder COI, was highlighted [33]. Currently, the most used molecular markers in Heterobranchia systematics, both at species and genus taxonomic levels, are the mitochondrial gene regions of the cytochrome oxidase subunit 1 (COI) and 16S rRNA (16S) and the nuclear H3 DNA region. In fact, previous studies showed that the two mitochondrial markers are well suited to study closely related nudibranch species [34–37], while nuclear H3 revealed to be useful only at higher taxonomic levels due to its high conservation and slower mutation rate [38].

Hitherto, none of the previous molecular studies on *Doriopsilla* genus included Mediterranean *D. areolata* specimens. This is quite surprising being *D. areolata* the type-taxon of the genus and the Adriatic Sea (Mediterranean Sea) the type locality. This is a significant deficiency not only in terms of principle but also when we consider how many integrative systematic studies have recently shown that Mediterranean populations often are separated from the Atlantic ones with several cases of endemism and cryptic and new species [35,36,38–47]. Therefore, taking all these factors into consideration, the aims of the present study are as follows: (1) to molecularly compare the Mediterranean and Atlantic

*D. areolata* populations by adding an additional useful molecular marker, COI barcoder, on an extended dataset; (2) to investigate on the possible occurrence of cryptic diversity within *D. areolata* in the Mediterranean Sea; and (3) to define the range of variability of the color pattern characterizing *D. areolata*.

## 2. Materials and Methods

*Doriopsilla areolata* individuals included in the present study were hand collected in different regions by SCUBA diving between 2012 and 2021 along the coasts of the Mediterranean Sea and the Eastern Atlantic Ocean (Table 1). In particular, 16 samples were collected in the Mediterranean Sea: nine of them were from Italian coastal areas, three were sampled from Spain, one sample was from Croatia, one from Tunisia, and the remaining three samples came from France. Two samples were collected from the coast on Spain in the East Atlantic Ocean. Sequences from other extra Mediterranean specimens were obtained from GenBank. Collected samples were photographed in situ and in laboratory, preserved in 95% ethanol (EtOH) for upcoming molecular analyses, and deposited in the Department of Science of the Roma Tre University collection (Vouchers RM3_ID number).

**Table 1.** Species name, voucher code, sampling locality, and GenBank accession numbers of all the analyzed specimens are listed, together with the outgroups. In bold are the specimens examined in this study.

| Species | Voucher | Locality | Accession Numbers | | |
|---|---|---|---|---|---|
| | | | H3 | 16S | COI |
| *Doriopsilla albopunctata* | LACM:DISCO 11426 | White Point, California, USA (E-PAC) | - | - | MK550636 |
| | CPIC 00909 | Long Beach, California, USA (E-PAC) | - | KR002428 | KR002480 |
| | CPIC 00915 | Long Beach, California, USA (E-PAC) | KR002524 | KR002429 | KR002481 |
| | CPIC 00916 | Long Beach, California, USA (E-PAC) | - | KR002430 | KR002482 |
| | CPIC 00917 | Long Beach, California, USA (E-PAC) | KR002525 | - | KR002483 |
| | CPIC 00930 | Malibu, California, USA (E-PAC) | KR002527 | KR002431 | KR002485 |
| | CPIC 00932 | Malibu, California, USA (E-PAC) | KR002528 | KR002432 | KR002486 |
| | CPIC 01254 | Shell Beach, California, USA (E-PAC) | KR002535 | KR002440 | KR002494 |
| | CPIC 01255 | Shell Beach, California, USA (E-PAC) | KR002536 | KR002441 | KR002495 |
| | CPIC 01084 | Mission Bay, California, USA (E-PAC) | - | KR002442 | KR002496 |
| | CPIC 01083 | Redondo Beach, California, USA (E-PAC) | - | KR002443 | KR002497 |
| | CPIC 00918 | Long Beach, California, USA (E-PAC) | KR002526 | - | KR002484 |
| | CPIC 01239 | Carpinteria, California, USA (E-PAC) | KR002534 | KR002439 | KR002493 |
| | CPIC 00987 | Carpinteria, California, USA (E-PAC) | KR002529 | KR002434 | KR002488 |
| | CPIC 00986 | Carpinteria, California, USA (E-PAC) | - | KR002433 | KR002487 |
| | LACM:3420 | Carpinteria, California, USA (E-PAC) | KR002530 | KR002435 | KR002489 |
| | - | Newport Beach, California, USA (E-PAC) | KR002531 | KR002436 | KR002490 |
| | - | Newport Beach, California, USA (E-PAC) | KR002532 | KR002437 | KR002491 |
| | - | Newport Beach, California, USA (E-PAC) | KR002533 | KR002438 | KR002492 |
| *Doriopsilla areolata* | **RM3_109** | **Hvar, Croatia (MED)** | **ON209460** | **ON229526** | **ON211997** |
| | **RM3_1169** | **Portopiccolo, Sistiana, Trieste, Italy (MED)** | **ON209466** | **ON229532** | **ON211996** |
| As *D. areolata* in GB | MNCN:15.05/23766 | Girona, Catalonia, Spain (MED) | KC171040 | KC171023 | - |
| | MNCN:15.05/23782 | Las Palmas, Gran Canaria, Spain (E-ATL) | KC171037 | KC171026 | - |
| | LACM:2001-10.3 | Cadiz, Andalusia, Spain (E-ATL) | KC171035 | KC171024 | - |
| | LACM:2001-10.4 | Cadiz, Andalusia, Spain (E-ATL) | KC171036 | KC171025 | - |
| | MNCN:15.05/23784 | Angola (E-ATL) | KC171039 | KC171033 | - |
| | MNCN:15.05/23789 | Angola (E-ATL) | KC171038 | KC171031 | - |
| *Doriopsilla bertschi* | CPIC 00976 | Bahia de los Angeles, Baja California, Mexico (E-PAC) | KR002551 | - | KR002515 |
| | CPIC 01058 | Bahia de los Angeles, Baja California, Mexico (E-PAC) | KR002552 | KR002462 | KR002517 |
| | CPIC 01059 | Bahia de los Angeles, Baja California, Mexico (E-PAC) | - | KR002463 | KR002518 |
| | LACM:140785 | Bahia de los Angeles, Baja California, Mexico (E-PAC) | KR002561 | KR002471 | KR002519 |

**Table 1.** *Cont.*

| Species | Voucher | Locality | Accession Numbers | | |
|---|---|---|---|---|---|
| | | | H3 | 16S | COI |
| *Doriopsilla davebehrensi* | LACM:3421 | Bahia de los Angeles, Baja California, Mexico (E-PAC) | KR002553 | KR002464 | - |
| | CPIC 01038 | Bahia de los Angeles, Baja California, Mexico (E-PAC) | KR002564 | KR002475 | KR002520 |
| | LACM:3419 | Bahia de los Angeles, Baja California, Mexico (E-PAC) | KR002565 | KR002476 | KR002521 |
| | LACM:76-5.6 | Bahia de los Angeles, Baja California, Mexico (E-PAC) | KR002566 | KR002478 | - |
| | - | Newport Beach, California, USA (E-PAC) | - | KR002477 | KR002522 |
| *Doriopsilla fulva* | CPIC 00933 | Malibu, California, USA (E-PAC) | KR002537 | KR002444 | KR002498 |
| | CPIC 00934 | Malibu, California, USA (E-PAC) | KR002538 | KR002445 | KR002499 |
| | CPIC 00936 | Malibu, California, USA (E-PAC) | KR002539 | KR002446 | KR002500 |
| | CPIC 00937 | Malibu, California, USA (E-PAC) | - | KR002447 | KR002501 |
| | CPIC 01240 | Mendocino, California, USA (E-PAC) | KR002541 | KR002449 | KR002503 |
| | CPIC 01022 | Palos Verdes, California, USA (E-PAC) | KR002540 | KR002448 | KR002502 |
| *Doriopsilla gemela* | CPIC 00923 | Malibu, California, USA (E-PAC) | KR002542 | KR002450 | - |
| | CPIC 00924 | Malibu, California, USA (E-PAC) | KR002543 | KR002451 | KR002504 |
| | CPIC 00931 | Malibu, California, USA (E-PAC) | - | KR002452 | KR002505 |
| | CPIC 00938 | Malibu, California, USA (E-PAC) | KR002544 | KR002453 | KR002506 |
| | CPIC 00939 | Malibu, California, USA (E-PAC) | KR002545 | KR002454 | KR002507 |
| | CPIC 00978 | Carpinteria, California, USA (E-PAC) | KR002546 | KR002455 | KR002508 |
| | CPIC 00979 | Carpinteria, California, USA (E-PAC) | KR002547 | KR002456 | KR002509 |
| | CPIC 00980 | Carpinteria, California, USA (E-PAC) | - | KR002457 | KR002510 |
| | CPIC 00981 | Carpinteria, California, USA (E-PAC) | - | KR002458 | KR002511 |
| | CPIC 00982 | Carpinteria, California, USA (E-PAC) | KR002548 | KR002459 | KR002512 |
| | CPIC 00983 | Carpinteria, California, USA (E-PAC) | KR002549 | KR002460 | KR002513 |
| | CPIC 00984 | Carpinteria, California, USA (E-PAC) | KR002550 | KR002461 | KR002514 |
| *Doriopsilla janaina* | CPIC 00590 | Peru (E-PAC) | KC171034 | KC171022 | - |
| *Doriopsilla miniata* | CAS:IZ:176370 | South Africa (E-ATL) | KC171043 | KC171030 | - |
| | CAS:IZ:176418 | Western Cape, South Africa (E-ATL) | KC171041 | KC171028 | - |
| | CAS:IZ:176933 | Western Cape, South Africa (E-ATL) | KC171042 | KC171029 | - |
| *D. pelseneeri* | **RM3_177** | **Tarifa, Andalusia, Spain (E-ATL)** | **ON209459** | **ON229525** | **ON211995** |
| As *D. areolata* in GB | - | Cadiz, Andalusia, Spain (E-ATL) | - | AJ225186 | AJ223262 |
| | - | Berlengas, Portugal (E-ATL) | - | KT820536 | KT833266 |
| | - | Berlengas, Portugal (E-ATL) | - | - | KT833267 |
| *D. rarispinosa* | **RM3_768** | **Tavolara Island, Sardinia, Italy (MED)** | **ON209461** | **ON229527** | **ON211998** |
| | **RM3_770** | **Tavolara Island, Sardinia, Italy (MED)** | **ON209462** | **ON229528** | **ON211999** |
| | **RM3_771** | **Tavolara Island, Sardinia, Italy (MED)** | **ON209463** | **ON229529** | **ON212000** |
| | **RM3_772** | **Tavolara Island, Sardinia, Italy (MED)** | **ON209464** | **ON229530** | **ON212001** |
| | **RM3_498** | **Kerkennah, Tunisia (MED)** | **ON209465** | **ON229531** | **ON212002** |
| | **RM3_1278** | **Golfo di Olbia, Sardinia, Italy (MED)** | **ON209467** | **ON229533** | **ON212003** |
| | **RM3_1280** | **Golfo di Olbia, Sardinia, Italy (MED)** | **ON209468** | **ON229534** | **ON212004** |
| | **RM3_1282** | **Golfo di Olbia, Sardinia, Italy (MED)** | **ON209469** | **ON229535** | **ON212005** |
| | **RM3_1286** | **Golfo di Olbia, Sardinia, Italy (MED)** | **ON209470** | **ON229536** | **ON212006** |
| | **RM3_2187** | **Palavas-les-flots, Occitanie, France (MED)** | **ON209471** | **ON229537** | **ON212007** |
| | **RM3_2188** | **Palavas-les-flots, Occitanie, France (MED)** | **ON209472** | **ON229538** | **ON212008** |
| | **RM3_2189** | **Palavas-les-flots, Occitanie, France (MED)** | **ON209473** | **ON229539** | **-** |
| | **RM3_2190** | **L'Escala, Girona, Catalonia, Spain (MED)** | **ON209474** | **ON229540** | **ON212009** |
| | **RM3_2191** | **L'Escala, Girona, Catalonia, Spain (MED)** | **ON209475** | **ON229541** | **ON212010** |
| *Doriopsilla spaldingi* | CPIC 00908 | San Pedro, California, USA (E-PAC) | KR002523 | KR002427 | KR002479 |
| *Doriopsilla* sp. 1 | **RM3_194** | **Tarifa, Andalusia, Spain (E-ATL)** | **ON209458** | **ON229524** | **ON211994** |
| *Felimare tricolor* | BAU 20547 | Secche di Tor Paterno MPA, Lazio, Italy (MED) | MK474153 | LN715193 | LN715211 |
| *Phyllidia coelestis* | CAS:IZ:190982 | Kranket Island, Madang Prov., Papua New Guinea (W-PAC) | - | MF958279 | MF958412 |
| | Phco18LS1 | Lembeh, North Sulawesi, Strait Indonesia (IWP) | - | MK852557 | MK911039 |
| *Phyllidia flava* | **RM3_546** | **Giglio Island, Tuscany, Italy (MED)** | **ON209476** | **-** | **ON212011** |

*Molecular Analyses*

DNA was extracted from the body tissues using the 'salting out' procedure [48]. First, a small piece of tissue was cut from the tail and placed in a tube where it was heated for one hour at 40 °C. In the following step, 430 µL of Cell Lysis Buffer and 20 µL of Proteinase K were added to the dried tissue. The samples were then incubated in a thermoblock overnight at a 56 °C. Next, the samples were vortexed and centrifuged at 13,200 rpm for 10 min. After this first centrifugation, the liquid supernatant was carefully pipetted into new tubes. Afterwards, 160 µL of NaCl 5 M was added to the samples, and these were gently vortexed and centrifuged for 10 min at 13,200 rpm. The supernatant was carefully taken and placed into the final tubes, and 500 µL of isopropanol was added. Next, the samples were gently vortexed and centrifuged under the same conditions used in the previous steps and finally the supernatant was discarded, leaving the DNA pellet adhering to the wall of the tubes. One ml of 80% EtOH was added, and the tubes were centrifuged for the last time for 10 min at 13,200 rpm. The supernatant was carefully discarded again, and the samples were left to dry for 1–2 h at room temperature. Finally, dried samples were diluted with the 60–100 µL of double distilled $H_2O$. Two different mitochondrial gene regions, COI and 16S, and the nuclear H3 were amplified. The universal primers LCO1490 and HCO2198 [49] and 16Sar-L and 16Sbr-H [50] were used for the COI and 16S mitochondrial markers, respectively, while H3AD-F and H3BD-R universal primers [51] were used for nuclear H3. The temperature profile for the PCR reactions was the same for the three molecular markers at the beginning of an initial denaturation step at 94 °C, which lasted 5 min. This step was followed by 35 cycles consisting of 30 s. at 94 °C for the denaturation step, 60 s. at an annealing temperature of 46–50 °C and 60 s. at an elongation temperature of 72 °C. After this cycle, the temperature was held for another 7 min. at 72 °C. Once all these steps were completed, the entire reaction was cooled down to a temperature of 10 °C. The PCR reaction mix has a final volume of 20 µL and consisted of 14.6 µL of $dH_2O$, 4.0 µL of 5x FIREPol Mastermix (5x Reaction buffer (0.4 M Tris-HCl, 0.1 M $(NH_4)_2SO_4$, 0.1% *w/v* Tween-20], 12.5 mM $MgCl_2$, 1 mM dNTP), 0.2 µL of each forward and reverse primers (20 µM), and 1.0 µL DNA. The quality of all obtained PCR products was controlled on 1.2% agarose gels. Samples were sequenced by Macrogen Europe (1105 Amsterdam, The Netherlands). Before the sequences were used for the alignment, they were controlled with the Basic Local Alignment Search Tool (BLAST) to exclude possible contamination. Sequences were aligned together with GenBank sequences using the Muscle algorithm implemented in MEGA 6.0 [52]. Four different alignments were generated, and three single-gene dataset (COI, 16S, and H3) and one with the three genes were concatenated and partitioned (ConcDNA). Primer regions were always removed from the final alignments. In the case of the 16S alignment, it was proofread with Gblocks 0.91b [53,54], allowing less strict flanking positions as the less stringent setting selection, to remove the hyper-divergent regions. The best-fitting evolutionary model for each of the four datasets (three single genes and one concatenated and partitioned) was determined by using JModelTest version 2.1.10 under the BIC model [55]. To generate the concatenated and partitioned dataset, the program DnaSP 6.12.03 [56] was used. The mean *p*-distances between groups were calculated using MEGA 6.0 [52].

Different types of species delimitation analyses were carried out. In particular, we used ASAP (available at http://wwwabi.snv.jussieu.fr/public/abgd/) (accessed on 6 February 2022) to detect the barcode gap in the distribution of pairwise distances calculated on the COI sequence alignment [57,58]. ASAP analysis was performed on the in-group dataset using the Kimura Two Parameter (K2P) genetic distance and the default settings parameters. The Species Identifier program [59] was used to calculate maximum intraspecific and minimum interspecific distances (*p*-distance) and for clustering sequences based on pairwise distances. To assess the number of putative species in our COI DNA dataset, we used the Poisson Tree Processes model as implemented in the PTP web server [60] applied on the Bayesian tree. This species delimitation method outperforms other methods based on single-locus molecular phylogenies [60].

Two different phylogenetic analyses were carried out. Bayesian inference analysis (BI) was performed using the program MrBayes (v. 3.2.6) [61] by applying a Bayesian posterior likelihood methodology. Each of the four runs were conducted with four MCMCs (Markov Chain–Monte Carlo) for five million generations, a sample frequency of one tree per 1000 generations, and a burn-in of 25%. The maximum likelihood analysis was performed in raxmlGUI 1.5b2 [62], a graphical front-end for RAxML 8.2.1 [63], with 100 independent ML searches and 1000 bootstrap replicates. *Felimare tricolor* Cantraine, 1835 species was used as the outgroup for this analysis.

## 3. Results

A total of 53 *Doriopsilla* sequences were obtained from 18 specimens (Table 1), with 47 sequences derived from 16 individuals sampled in the Mediterranean Sea and six from two Eastern Atlantic Ocean individuals. Furthermore, 157 sequences from GenBank, including all available *Doriopsilla* species, were added to the final dataset, leading to a total of 209 sequences (Table 1).

Results from the mean *p*-distances (COI) calculated between the groups here investigated are reported in Table 2.

**Table 2.** Mean *p*-distances of the COI mitochondrial marker between *Doriopsilla* species and other related genera analyzed in the present study.

| | 1. | 2. | 3. | 4. | 5. | 6. | 7. | 8. | 9. | 10. | 11. | 12. |
|---|---|---|---|---|---|---|---|---|---|---|---|---|
| **1. *D. albopunctata*** | - | | | | | | | | | | | |
| **2. *D. areolata*** | 0.18 | - | | | | | | | | | | |
| **3. *D. bertschi*** | 0.16 | 0.16 | - | | | | | | | | | |
| **4. *D. davebehrensi*** | 0.08 | 0.18 | 0.16 | - | | | | | | | | |
| **5. *D. fulva*** | 0.09 | 0.17 | 0.15 | 0.08 | - | | | | | | | |
| **6. *D. gemela*** | 0.15 | 0.16 | 0.10 | 0.15 | 0.15 | - | | | | | | |
| **7. *D. pelseneeri*** | 0.17 | 0.03 | 0.16 | 0.18 | 0.16 | 0.17 | - | | | | | |
| **8. *D. rarispinosa*** | 0.19 | 0.06 | 0.15 | 0.17 | 0.17 | 0.16 | 0.06 | - | | | | |
| **9. *D. spaldingi*** | 0.18 | 0.18 | 0.15 | 0.16 | 0.17 | 0.15 | 0.18 | 0.17 | - | | | |
| **10. *Doriopsilla* sp. 1** | 0.21 | 0.09 | 0.17 | 0.20 | 0.20 | 0.16 | 0.09 | 0.09 | 0.18 | - | | |
| **11. *Phyllidia* spp.** | 0.18 | 0.18 | 0.17 | 0.18 | 0.17 | 0.16 | 0.18 | 0.17 | 0.17 | 0.19 | - | |
| **12. *Felimare tricolor*** | 0.22 | 0.18 | 0.19 | 0.22 | 0.21 | 0.19 | 0.17 | 0.17 | 0.19 | 0.19 | 0.19 | |

All resulting trees were congruent with each other, differing only in the ability to resolve phylogenetic relationships at different taxonomic levels.

Bayesian inference (BI) and maximum likelihood (ML) analyses were generated from the single COI gene dataset consisting of 68 sequences with a length of 633 bp from 64 *Doriopsilla* and three *Phyllidia* specimens (Figure 1).

TPM1uf + I + G resulted to be the best evolutionary model for this single gene dataset. A clade formed by *Doriopsilla* and *Phyllidia* specimens was strongly supported by BI analysis and was clearly separated from the basal outgroup (Bayesian posterior probability (BPP) = 1, Bootstrap (BP) = <50). The monophyly of the genus *Doriopsilla* was strongly supported only for the Bayesian analysis (BPP = 1, BP = 68). Inside the *Doriopsilla* clade, the *D. spaldingi* Valdés and Behrens, 1998, specimen obtained from GenBank had no statistical support. Within the rest *Doriopsilla* clade, *D. albopunctata* Cooper, 1863, *D. fulva* MacFarland, 1905, and *D. davebehrensi* Hoover et al., 2015, were grouped in a well-supported monophyletic group (BPP = 1, BP = 96). Within this group, the monophyly of these three species was well supported with BPP = 1 and BP = 96 for *D. davebehrensi* and *D. fulva* and with BPP = 1 and BP = 95 for *D. albopunctata*, respectively. Another clade strongly supported by BI includes *D. gemela* Gosliner et al., 1999, and *D. bertschi* Hoover et al., 2015 (BPP = 1, BP = <50). Within this group, *D. bertschi* and *D. gemela* are each monophyletic with strong statistical support at BI and moderately supported for the ML of the latter species (BPP = 1 and BP = 100; and BPP = 1 and BP = 70, respectively). Considering the Mediterranean/Atlantic species group, they form a well-supported clade (BPP = 1, BP = 88), which are in turn divided into two

groups: one grouping all the Mediterranean specimens previously identified as *D. areolata* and called from now on as *D. rarispinosa* (BPP = 1, BP = 95), and the second consisting in *D. pelseneeri*, *D. areolata*, and *Doriopsilla* sp. 1. *Doriopsilla areolata* (BPP = 0.99, BP = 100) and *D. pelseneeri* (BPP = 1, BP = 90) were grouped in a monophyletic clade (BPP = 0.95, BP = <50) that is a sister to *Doriopsilla* sp.1 with low support (BPP = 0.66, BP = <50).

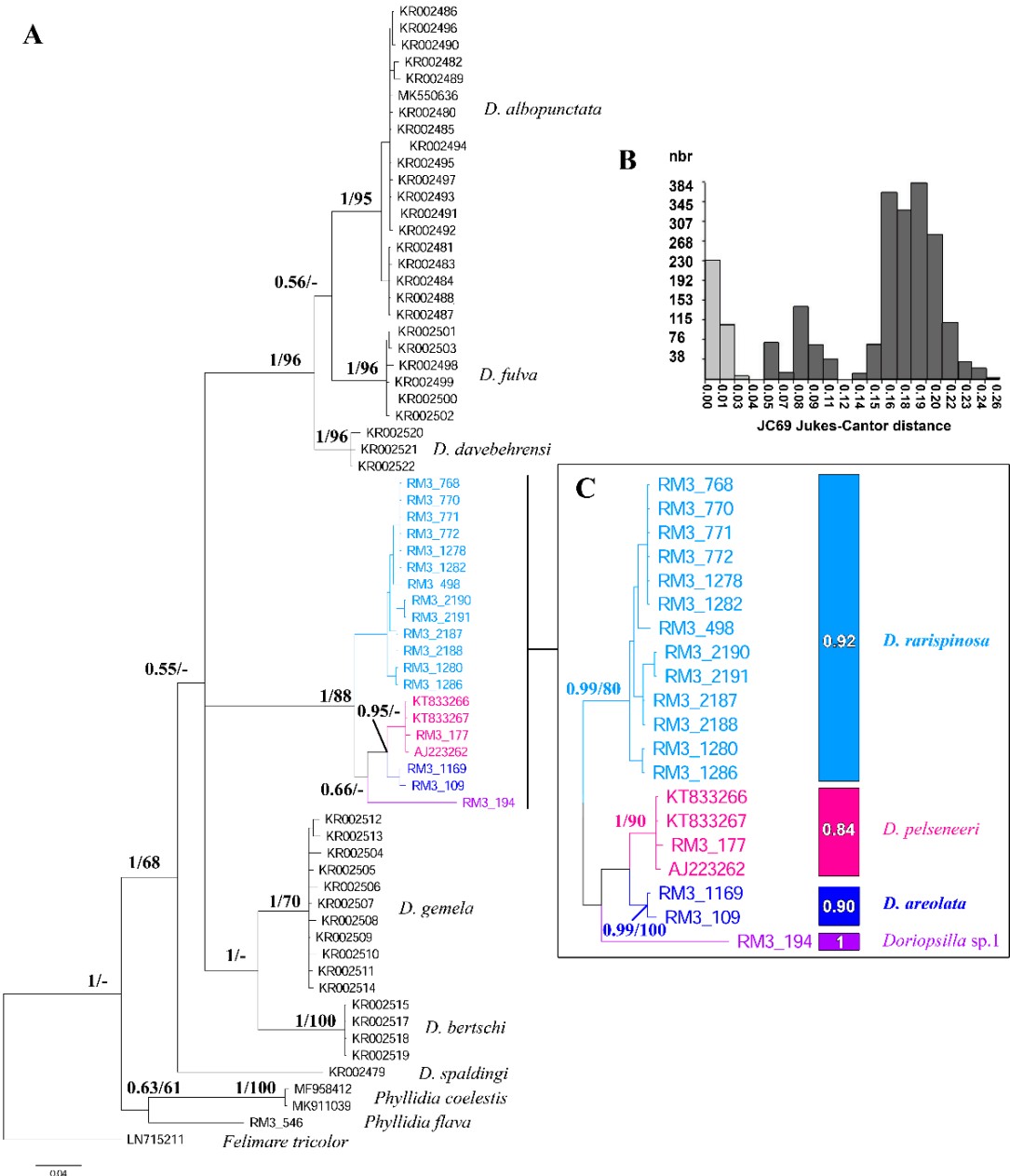

**Figure 1.** Bayesian inference tree based on the COI sequence dataset along with results from species-delimitation analyses. (**A**) Bayesian COI tree. Numbers at nodes indicate Bayesian Posterior probability (BPP; left) and bootstrap support from the maximum likelihood analysis (BP; right). BPP < 0.50 and BP < 50% are not reported. (**B**) The histogram shows the distribution of the pairwise genetic distances (JC69) in intraspecific (left, light grey) and interspecific (right, dark grey) comparisons. (**C**) Species delimitation analyses on the Mediterranean *Doriopsilla* spp. The colored rectangles show the results from the ASAP analysis with the Bayesian support values from the PTP reported inside each rectangle. The '-' symbol indicates unsupported values.

The concatenated and partitioned (ConcDNA) dataset consisted of 79 sequences with a bp length of 1315 obtained from 74 specimens belonging to *Doriopsilla*, three belonging to *Phyllidia*, and the outgroup *Felimare tricolor* (Figure 2).

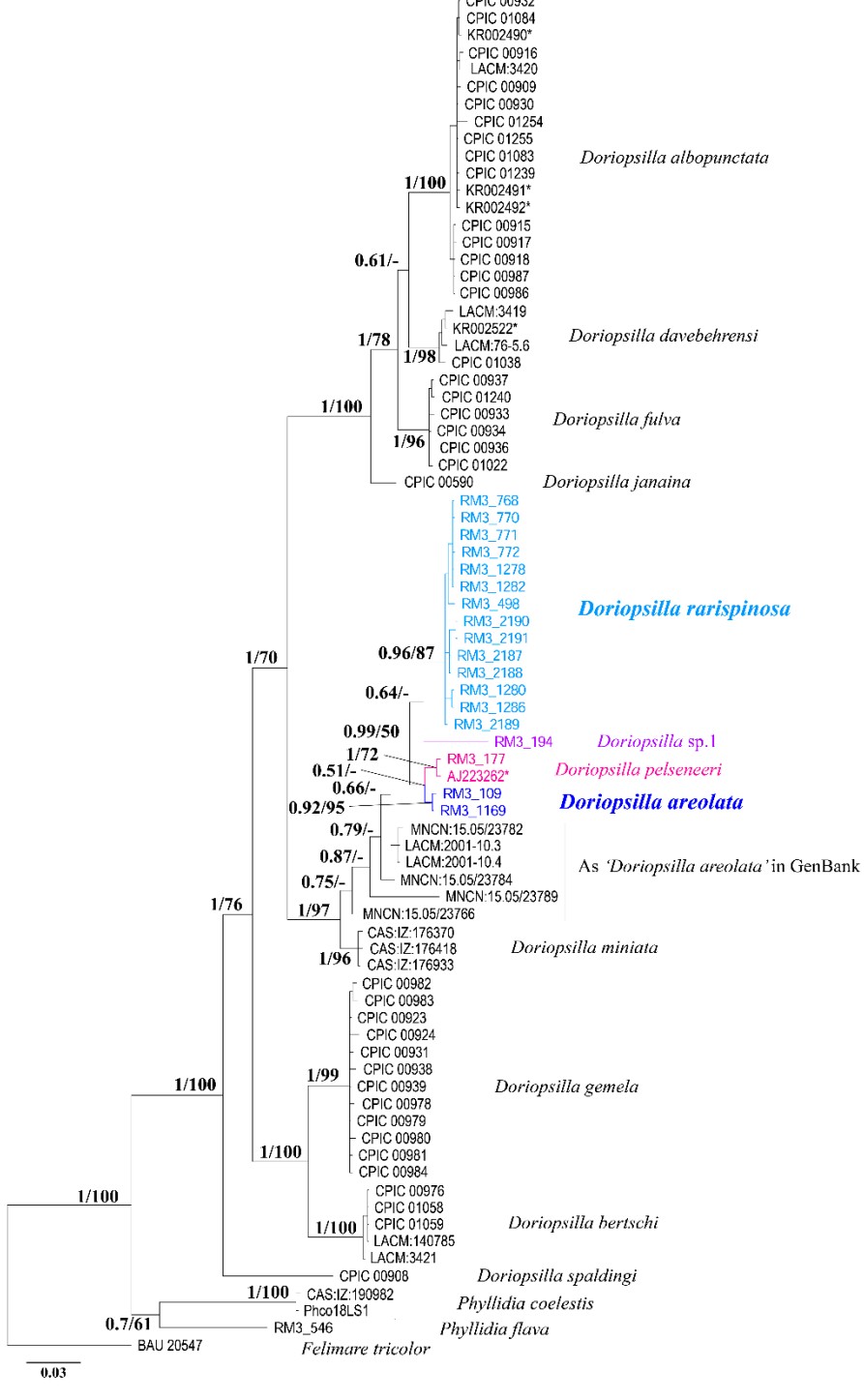

**Figure 2.** Bayesian phylogenetic tree based on the ConcDNA dataset (H3, 16S, COI). Bayesian Posterior probability (BPP; left) and bootstrap support from the maximum likelihood analysis (BP; right) are indicated at each node. The '-' symbol indicates unsupported values.

The evolutionary model used for COI and H3 calculations was HKY + G, while the TPM3uf + G was the one selected for the 16S. *Doriopsilla* and *Phyllidia* formed a clade that is well supported and separated from the basal outgroup (BPP = 1, BP = 100). All *Doriopsilla* species formed a strongly supported monophyletic clade (BPP = 1, BP = 100),

with *D. spaldingi* as the basal species, which is a sister to all the other *Doriopsilla* spp. The remaining eight *Doriopsilla* were grouped in a monophyletic clade (BPP = 1, BP = 76) and, in turn, divided in two monophyletic clades. *Doriopsilla bertschi* (BPP = 1, BP = 100) and *D. gemela* (BPP = 1, BP = 99) formed one of these clades (BPP = 1, BP = 100). The second one is composed by the remaining *Doriopsilla* species (BPP = 1, BP =70). This second monophyletic clade was divided into two big clades. One monophyletic well-supported group (BPP = 1, BP = 100) comprises the *D. janaina* sister to a clade with *D. fulva* (BPP = 1, BP = 96), *D. davebehrensi* (BPP = 1, BP = 98) and *D. albopunctata* (BPP = 1, BP = 100). *Doriopsilla fulva* is the sister (BPP = 1, BP = 78) to the two latter species. The second big group (BPP = 1, BP = 97) is composed by the Mediterranean/Atlantic *Doriopsilla* spp. with *D. miniata* (BPP = 1, BP = 96) as the sister species to all the remaining ones. A group of non-supported sequences named as '*D. areolata*' is the sister to a monophyletic clade (BPP = 0.99, BP = 50), which grouped *D. areolata* (BPP = 0.92, BP = 95), *D. pelseneeri* (BPP = 1, BP = 72), *D. rarispinosa* (BPP = 0.96, BP =87), and *Doriopsilla* sp.1.

The results of the phylogenetic analyses showed that the genus *Doriopsilla* formed a monophyletic clade separated from the outgroup. Furthermore, COI as well as ConcDNA dataset revealed *D. areolata* as a complex of at least three different species (Figures 1 and 2). Based on this species division, the Adriatic population is, henceforth, listed as the *bona fide D. areolata*, since Bergh [27] used this population for the original description. The Western-central Mediterranean population formed a monophyletic clade that is from now on referred to as *D. rarispinosa*. The reassumption of this name for this clade is supported by the following criteria: (I) The original description and the subsequent redescription (Perrone 1989) show a high morphological correspondence with the specimens here examined. (II) *D. reticulata* (the first species whose description corresponds to our specimens and subsequently considered synonym of *D. areolata*) cannot be used, as already specified, as a junior homonymous of *D. reticulata* Quoy & Gaimard, 1832 because it is invalid. (III) *D. pusilla* described in the same year as *D. rarispinosa* was excluded considering that Pruvot-Fol herself assessed it to be of an uncertain genus, that it lacks a description allowing its definition, and probably the specimen, which is no longer available, was in the juvenile stage (3 mm). (IV) The third available name, *D. fedalae*, has been excluded, being an Atlantic species described in 1953, after *D. rarispinosa*, and it is quite different in external morphology. It is possible that this taxon may be used to name the Atlantic *Doriopsilla* species, which is up until now erroneously named as *D. areolata*. However, Atlantic populations need an in-depth systematic study that proceeds beyond the aims of the present work. Finally, an additional *Doriopsilla* species was revealed in the Atlantic Ocean; based on the results of COI and ConcDNA analyses, this specimen should be regarded as a separate species and is here provisionally reported as *Doriopsilla* sp.1.

## 4. Discussion

The range-wide sampling along the Mediterranean and Atlantic areas allowed revealing cryptic diversity within *D. areolata* species (Figure 3).

In fact, even if the diversity of the Mediterranean nudibranchs is still far from being comprehensively unveiled, several steps forward have been made thanks to molecular methods [38]. The morphological and anatomical identification of nudibranch species is based on characteristics that are quite variable, difficult to compare, and not always present [38]. The radula, the hard structure part of the buccal apparatus typically used in nudibranchs for feeding, is lacking in the *Doriopsilla* species, failing one of the most important and diagnostic morphological characters. Therefore, particularly in the case of the Dendrodorididae, morphological investigations could leave room for errors and confusion in species identification. On the contrary, molecular identification methods seem to be the most powerful tool to reveal taxonomic misinterpretations made in the past and to highlight the presence of cryptic species across different marine animal groups, especially in the Mediterranean Sea where several cases of cryptic species and cases of endemism are continuously reported [38,43–46,64].

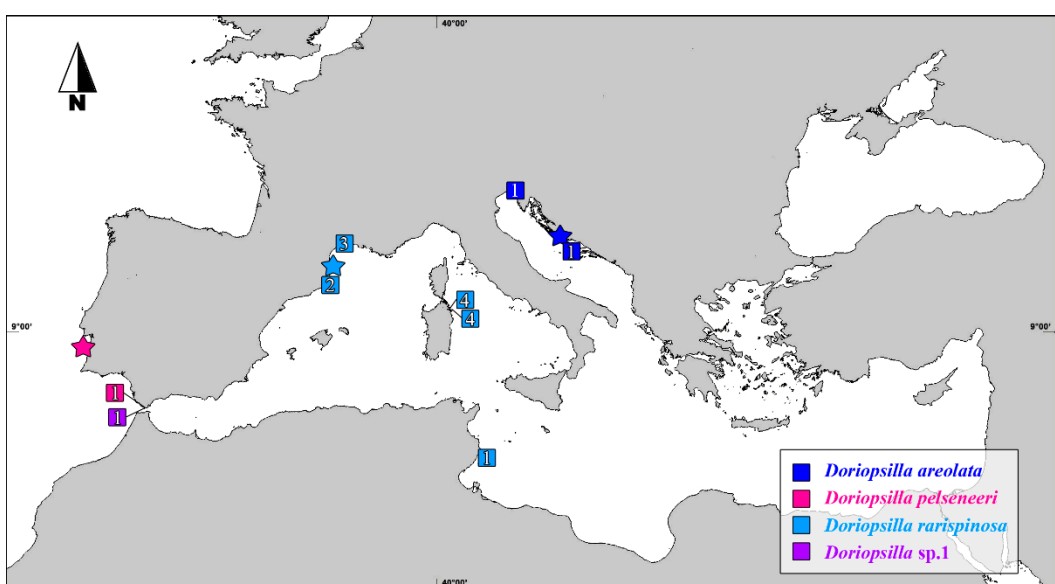

**Figure 3.** Map showing the type localities (stars) and the provenance of the Atlantic/Mediterranean *Doriopsilla* specimens included in the present study (squares). The numbers within the squares refer to the number of individuals collected at each site.

In the present work, molecular investigations implemented with the COI barcoder marker above the already used 16S and H3 markers and carried out on an extended dataset helped to unveil, as hypothesized, the presence of at least three different species under the *D. areolata* complex, two of which are *D. areolata* and *D. rarispinosa*, which are endemic to the Mediterranean Sea (Figures 1–3).

Interestingly, no attention has been paid before to the possible separation between Mediterranean and Atlantic populations of *D. areolata* (e.g., in [5,15,28,33]). In fact, until now, only the populations from the Eastern Atlantic and Caribbean Sea have been studied but without including specimens from the type localities (i.e., the Adriatic Sea), thus ignoring an important and reference point for this species (e.g., [15,28]). Therefore, by filling the gap, this study shed some light on the Mediterranean population of *D. areolata*, revealing a new case of cryptic diversity. Results from phylogenetic analyses (considering single gene datasets and concatenated and partitioned data) were congruent with each other considering *D. areolata*, as previously conceived, as a complex of at least three distinct phylogenetic lineages.

*Doriopsilla areolata* specimens from the Adriatic Sea (RM3_109 and RM3_1169) formed one of these well-separated and monophyletic lineages and appeared to be more related to Atlantic *D. pelseneeri* than to the other Mediterranean species (Figures 1 and 2). *Doriopsilla areolata* was described by Bergh in 1880 based on two specimens from Hvar Island in the Adriatic Sea. He described the ground coloration as light yellow–gray, with a reddish–brownish spot on the back (due to the peritoneum and viscera). Furthermore, he described that all over the mantle and up to the edge, there were thin whitish lines that branched and anastomosed to form an irregular grid with large and small meshes of different shapes. Bergh's description is consistent with the morphology of the specimens sampled from the Adriatic Sea and here reported (Figure 4A,B).

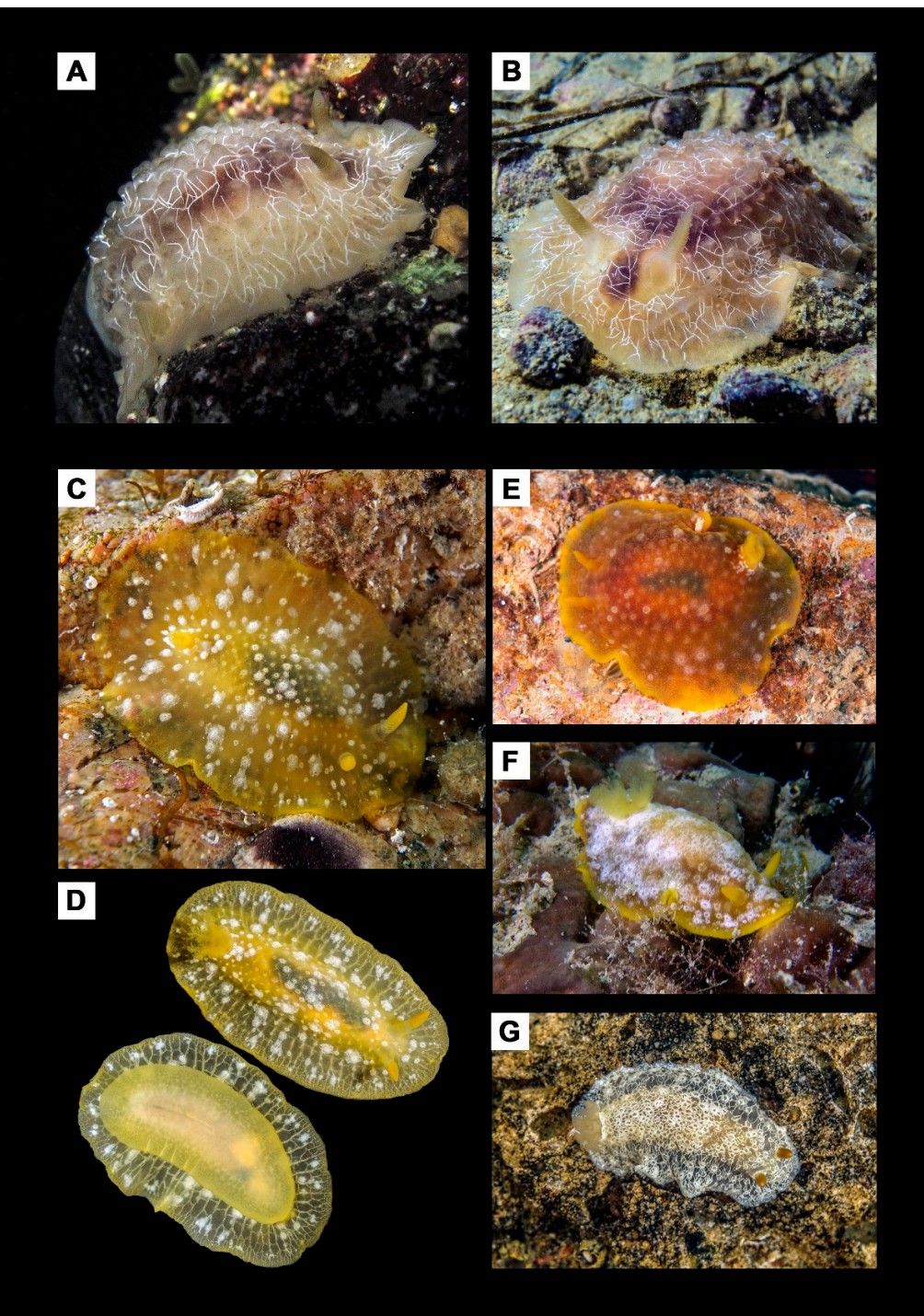

**Figure 4.** Mediterranean *Doriopsilla* species. (**A**,**B**) *Doriopsilla areolata* from the Adriatic Sea (in **B**, specimen with Voucher RM3_1169). *Doriopsilla rarispinosa* individuals from (**C**,**D**) Sardinia (Italy) (Voucher RM3_1278), (**E**) Catalonia (Spain) (Voucher RM3_2190), (**F**) Occitanie, France (Voucher RM3_2189), and (**G**) Tunisia (Voucher RM3_498).

The ground coloration was very light yellowish almost grayish transparent, and small white dots could be observed all over the body. On the dorsum, reddish-dark spots were present which originate from the organs lying underneath the epithelium. The dorsum was covered with small transparent tubercles, which could reach the mantle. An irregular grid pattern formed of prominent thin white lines could be observed on the dorsum and mantle. Since molecular evidence presented in this study indicates that *D. areolata* from the

Adriatic Sea forms a separate species and considering that the first description of Bergh [27] was based on individuals from the same geographic area, it could be concluded that the species from the Adriatic Sea (the type locality of the species) must be considered, from now on, as the *bona fide D. areolata*. Interestingly *D. areolata* species seems to be geographically restricted to the Adriatic Sea since the only *Doriopsilla* reported to date from the nearby Ionian Sea is *D. rarispinosa* [65,66].

The second monophyletic clade resulting from the molecular investigation reported here corresponds to *D. rarispinosa* Pruvot-Fol, 1951, which in fact could not be considered anymore as the synonym of *D. areolata* and was reinstated here as a valid species name according to the principle of priority of the International Code of Zoological Nomenclature (ICZN Art. 23.3.5). This species was originally described from Banyuls-sur-Mer, France, Mediterranean Sea [67], and this locality is in line with the Western-central Mediterranean distribution we observed (Figure 3). This species shows a variable body color pattern from specimens displaying a yellow–orange background (Figure 4C–E) to others characterized by a strong dominance of the white color (Figure 4F). The Western Mediterranean populations correspond well with Pruvot-Fol's description, the redescription by Perrone (1986) [68] and the image reported by Cattaneo-Vietti [69] (pag. 229 Figure 8). This correspondence with our samples is well represented in Figure 4D. It is noticeable that the coloration intensity of tubercles and lines and their texture do not change along a geographic gradient. In the middle of the dorsum, the underlying organs are visible as a dark grayish spot. It should also be noted that *D. rarispinosa* is the first available name, as *Doris reticulata* Schultz in Philippi, 1836 (despite its original description from a specimen collected in Palermo corresponds with our sample in Figure 4D), cannot be used, being an invalid junior synonym of *Doris reticulata* (Quoy & Gaimard, 1832). The Tunisian morphotype (Voucher RM3_498) (Figure 4G) shows a yellowish–white ground coloration of the dorsum. The mantle is whitish transparent, and in the dorsal area from the rhinophores to the gills, it assumes a yellowish color in correspondence with the internal organs. Flat tubercles, which are white bordered and transparent in the middle, are visible from the dorsum to the mantle edge. The most evident peculiarity of this morphotype is an irregular and distinctive web of fine white lines connected with the white spots on the dorsum and mantle, which are very different from the typical pattern of lines of *D. areolata*. Compared to the other *D. rarispinosa* morphotypes, the swarthy yellow–orange coloration of the rhinophores and their clearly rounded shape are outstanding and deserves further in-depth studies.

The third phylogenetic lineage reported here includes the Atlantic specimen (Voucher RM3_194) from Tarifa (Spain) that formed a well-supported clade in both COI and ConcDNA analyses (Figures 1 and 2), suggesting that it could be ascribed to another species. The color of the specimen was orange–yellowish, and different larger reddish–brown areas were visible on the dorsum because of the organs lying under the epithelium. Moreover, in this case, thin white lines forming an irregular grid were observed. Since this study mainly focused on the Mediterranean *D. areolata* species complex, we provisionally referred to it as *Doriopsilla* sp. 1 (Figure 5), but an in depth-study of this possible additional Atlantic/Mediterranean species is advisable. However, the results from the 16S single gene dataset analysis (not shown) revealed a sister relationship with an individual from Cape Verde (E-ATL) reported in GenBank as '*D. areolata*' (Voucher MNCN:15.05/23781 and 16S accession number KC171027), which deserves further investigations.

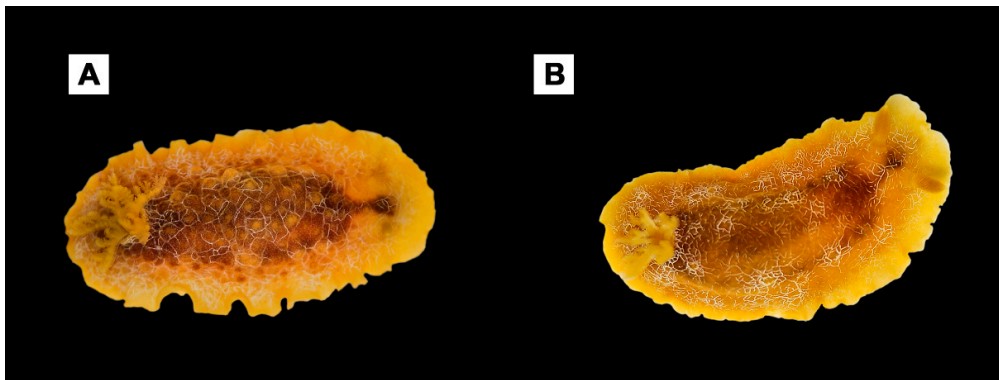

**Figure 5.** Atlantic *Doriopsilla* species analyzed here. (**A**) *Doriopsilla pelseneeri* (Voucher RM3_177) from Tarifa, Andalusia, Spain (E-ATL). (**B**) *Doriopsilla* sp.1 (Voucher RM3_194) from Tarifa, Andalusia, Spain (E-ATL).

The closer relation between *D. areolata* from the Adriatic Sea and the Atlantic species, in contrast to *D. rarispinosa* from the Western Mediterranean basin, was unexpected. As a possible explanation, the propagules of the Atlantic *Doriopsilla* could have entered and colonized the Adriatic Sea. A possible pathway for larval dispersion could be the unidirectional surface current called 'the Algerian current', which is known to trap and transport larvae [70,71]. The current starts in the Atlantic Ocean and moves eastwards through the Mediterranean along the Northern African coast, bypassing the entire Western-central Mediterranean basin until it reaches the northern coast of Tunisia near Kelibia [72]. At this point, the current splits into two main branches: The first reaches Lampedusa, and the second follows the Sicilian coast towards the Ionian Sea [73]. From there, a northward cyclonic flow could have allowed the entry of larvae through the Otranto Channel, which are then further distributed along the Balkan coasts towards the Northern Adriatic [74]. The relationships between Mediterranean *Doriopsilla* species could be also related to the post-Messinian flood, which ended the Messinian Salinity Crisis (6.0–5.3 Ma) [75,76]. During this event, the Mediterranean Sea was filled with Atlantic Ocean Sea water, and consequently, Atlantic species invaded the Mediterranean basin. This flood could have also led to the introduction of a possible *Doriopsilla* ancestor. This introduction could have caused the formation of two different phyletic lineages: the Adriatic lineage, which is more related to the ancestor, and another one, which is endemic to the Mediterranean. The close relation of the Adriatic lineage to Atlantic *Doriopsilla* species would be likely since the Adriatic Sea is cooler and less salty than the rest of the Mediterranean Sea, creating conditions more similar to those of the Atlantic Ocean [77]. The abiotic isolation and the fact that the Adriatic Sea is semi-enclosed showed, in different studies [78,79], that genetic isolation can occur and lead to speciation processes [78,79]. In order to effectively address these possible scenarios, an extended investigation on other possible Mediterranean cryptic species with the possibility to calibrate the analyses to investigate the ancestral areas, is strongly recommended.

Finally, the separation during the evolution of the group from the common ancestor occurred recently, and this could explain the lack of lineage sorting observed in the H3 nuclear marker. Therefore, for future perspectives, it could be considered to explore alternative and fast-evolving markers as, for example, nuclear ITS2 instead of H3 to obtain additional useful information at a lower taxonomic scale [38]. Cryptic diversity is an intriguing challenge, especially regarding the Mediterranean Heterobranchia fauna that deserves an integrative systematic approach due to the recently separated species and the close evolutionary history with relative Atlantic fauna.

## 5. Conclusions

The results of this study revealed that *D. areolata*, as previously conceived, is a complex of cryptic species that includes at least two different species, *D. areolata* and *D. rarispinosa*, endemic to the Mediterranean Sea. In fact, this latter species is valid, and *D. rarispinosa*

species name is here reinstated. Further phylogenetic analyses are still needed to investigate the possible additional species, provisionally reported here as *Doriopsilla* sp.1, whose occurrence may be searched both in the Eastern Atlantic Ocean, along the Spanish and Portuguese coasts, and in the Southern Mediterranean coasts of the Alboran Sea. Further research is also needed to solve the taxonomic problem of the Eastern Atlantic specimens, which is currently erroneously classified as *Doriopsilla areolata*.

**Author Contributions:** Conceptualization, G.F., E.T. and P.M.; methodology, G.F.; formal analysis, G.F., C.S. and E.T.; investigation, E.T., M.P., E.M. and P.G.; resources, E.T., M.P., E.M. and P.G.; data curation, G.F., C.S. and E.T.; writing—original draft preparation, G.F. and C.S.; writing—review and editing, G.F., E.T., M.P. and P.M.; supervision, P.M.; funding acquisition, G.F. All authors have read and agreed to the published version of the manuscript.

**Funding:** This research was funded by Italian Ministry of Education, University and Research MIUR, PON 2014–2020, grant number AIM 1848751-2, Linea 2, and the APC was funded by MIUR, PON 2014–2020.

**Institutional Review Board Statement:** Not applicable.

**Informed Consent Statement:** Not applicable.

**Data Availability Statement:** Not applicable.

**Acknowledgments:** This paper is dedicated to the late Barbara Camassa (Sistiana, Trieste), who collected samples from the Gulf of Trieste (Adriatic Sea) and whose contribution to the deepening of Mediterranean nudibranchs has yet to be fully developed. The authors wish to thank Stefano Piraino (Lecce, Italy) who helped improving the quality of this manuscript. Thanks is given to Lucas Cervera Currado (Cadiz, Spain) for his help in sampling the specimens from Tarifa. The collection of samples in the Marine Protected Area 'Tavolara-Punta Coda Cavallo' (Olbia, Italy) was authorized by the managing body, which is thanked here. GF wishes to thank the Scubalandia Team for technical underwater support. GF is supported by funds from the Italian Ministry of Education, University and Research (MIUR, PON 2014–2020, grant AIM 1848751-2, Linea 2).

**Conflicts of Interest:** The authors declare no conflict of interest. The funders had no role in the design of the study; in the collection, analyses, or interpretation of data; in the writing of the manuscript; or in the decision to publish the results.

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
