# Peer review of "The Sea Slug Doriopsilla areolata Bergh, 1880 (Mollusca, Gastropoda) in the Mediterranean Sea: Another Case of Cryptic Diversity"

_diversity, doi:10.3390/d14040297_

Round 1

Reviewer 1 Report

The manuscript represents a very interesting study of radula-less nudibranch Doriopsilla areolata in the Mediterranean Sea. Taking in account lack of the important diagnostic and taxonomical feature, the radula, observations on the genetic diversity and comparison with the morphological diversity are very important. 

Although I liked very much the manuscript in general, I have several comments and suggestions on improving the manuscript. 

  1. Better consistency between the introduction, the material and methods and the results sections. In the present form of the ms several paragraphs that clearly relate to the M&M sections, in fact are placed into the Results (see attached pdf). Also, in the Introduction the authors mentioned that D. rarispinosa is regarded as minor synonym of D. areolata, but further in the Results section this species name suddenly appears with no explanation. The explanations are given in the Discussion section, but again, other possible species names (D. fedalae, D. pusilla) are not discussed at all.
  2. Better consistency between molecular results and taxonomical decisions. Although the data obtained by authors indicates presence of four species in the D. areolata species complex, the addition of more sequences available from the GenBank indicate there are more species in this complex (or not?). Since those specimens from GenBank are also from the Mediterranean Sea, it seems important properly discuss this matter. I understand that the lack of resolution in the basal relationships within D. areolata species complex in ConcDNA dataset relates to lack of COI data for D. areolata specimens from GB. However, authors can provide clear analysis of 16S, for example by reconstructing of the haplotype network. 
  3. I cannot support designation of bootstrap support value of 50-70% as "high support". Please revise these criteria and provide them in the M&M section. And revise the entire Results section accordingly. 
  4. The level of molecular divergence within D. areolata species complex is not clear, please calculate p-distances for each marker and provide it as a table (at least supplementary one). 
  5. I liked very much the last part of discussion, and appreciate any thoughts on possible speciation drivers. However, I must stress that without calibrated molecular tree and the reconstruction of ancestral areas these thoughts look very speculative. Also, as in the above, these discussion does not take into account the presence of other possible cryptic species (GenBank data). 
  6. For my opinion, the manuscript would benefit from the clear statement on diagnostic morphological characters allowing clear identification of Doriopsilla species in the Mediterranean Sea. 

For other minor comments please see the PDF file attached. 

Reviewer 2 Report

I was excited by this paper, and the premises and data are strong, but the authors do not follow through. The first sections are fine bar typos and spelling mistakes. (The paper is now set in US English so take advantage of the automatic spell check.) The Discussion needs work as it jumps from topic to topic and is not coherent at all. I think the authors missed an opportunity to compare their sp. 1 with possible names in the literature. The figures are great! The references will need careful checking word by word as they are full of mistakes. I was not able to print so I have not checked that all refs are cited in both directions.
